# Rethinking Uncertainty in Deep Learning: Whether and How it Improves Robustness

## Abstract

Deep neural networks (DNNs) are known to be prone to adversarial attacks, for which many remedies are proposed. While adversarial training (AT) is regarded as the most robust defense, it suffers from poor performance both on clean examples and under other types of attacks, e.g. attacks with larger perturbations. Meanwhile, regularizers that encourage uncertain outputs, such as entropy maximization (EntM) and label smoothing (LS) can maintain accuracy on clean examples and improve performance under weak attacks, yet their ability to defend against strong attacks is still in doubt. In this paper, we revisit uncertainty promotion regularizers, including EntM and LS, in the field of adversarial learning. We show that EntM and LS alone provide robustness only under small perturbations. Contrarily, we show that uncertainty promotion regularizers complement AT in a principled manner, consistently improving performance on both clean examples and under various attacks, especially attacks with large perturbations. We further analyze how uncertainty promotion regularizers enhance the performance of AT from the perspective of Jacobian matrices $\nabla_X f(X; \theta)$, and find out that EntM effectively shrinks the norm of Jacobian matrices and hence promotes robustness.

## 1 Introduction

Deep neural networks (DNNs) have achieved great success in image recognition (Russakovsky et al., 2015), audio recognition (Graves & Jaitly, 2014), etc. However, as shown by (Szegedy et al., 2013), DNNs are vulnerable to *adversarial attacks*, where slightly perturbed *adversarial examples* can easily fool DNN classifiers. The ubiquitous existence of adversarial examples (Kurakin et al., 2016) casts doubts on real-world DNN applications. Therefore, many techniques are proposed to enhance the robustness of DNNs to adversarial attacks (Papernot et al., 2016; Kannan et al., 2018).

Regarding defenses to adversarial attacks, adversarial training (AT) (Goodfellow et al., 2014; Madry et al., 2017) is commonly recognized as the most effective defense. However, it illustrates poor performance on clean examples and under attacks stronger than what they are trained on (Song et al., 2018). Recent works (Rice et al., 2020) attribute such shortcomings to overfitting, but solutions, or even mitigation to them remain open problems. Meanwhile, many regularization techniques are proposed to improve robustness, among which two closely related regularizers, entropy maximization (EntM) and label smoothing (LS) show performance improvements under *weak* attacks (Pang et al., 2019; Shafahi et al., 2019) without compromising accuracy on clean examples. However, as stated by (Uesato et al., 2018; Athalye et al., 2018), strong and various kinds of attacks should be used to better approximate the *adversarial risk* of a defense. Therefore, it remains doubtful how they perform under stronger attacks *by themselves* and how their adversarial risk is.

One appealing property of both EntM and LS is that they both penalize over-confident predictions and promote prediction uncertainty (Pereyra et al., 2017; Szegedy et al., 2016). Therefore, both of them are used as regularizers that mitigate overfitting in multiple applications (Müller et al., 2019). However, it remains unclear whether EntM and LS can effectively regularize AT, where many other regularizers are ineffective (Rice et al., 2020). Therefore, in this paper, we perform an extensive empirical study on uncertainty promotion regularizers, i.e. entropy maximization and label smoothing, in the domain of adversarial machine learning. We carry out experiments on both regularizers, with and without AT, on multiple datasets and under various adversarial settings. We found out that, although neither EntM and LS are able to provide consistent robustness *by themselves*, both regular-

izers complement AT and improve its performances consistently. Specifically, we observe not only better accuracy on clean examples, but also better robustness, especially under attacks with larger perturbations (e.g. 9% improvement under perturbation $\varepsilon = 16/255$ for models trained with $8/255$ AT. See Table 1 and 3). In addition, we investigate the underlying mechanisms about how EntM and LS complement AT, and attribute the improvements to the shrunken norm of Jacobian matrix $\|\nabla_X f(X; \theta)\|$ ($\sim$10 times. See Sect. 6), which indicates better numerical stability and hence better adversarial robustness than AT.

To summarize, we make the following contributions:

1. We carry out an extensive empirical study about whether uncertainty promotion regularizers, i.e. entropy maximization (EntM) and label smoothing (LS), provide better robustness, and find out that while neither of them provides consistent robustness under strong attacks *alone*, both techniques serve as effective regularizers that improve the performance of AT *consistently* on multiple datasets, especially under large perturbations.

2. We provide further analysis on uncertainty promotion regularizers from the perspective of Jacobian matrices, and find out that by applying such regularizers, the norm of Jacobian matrix $\|\nabla_X f(X; \theta)\|$ is significantly shrunken, leading to better numerical stability and adversarial robustness.

## 2   Related Work and Discussions

**Adversarial Attacks and Defenses.** (Szegedy et al., 2013; Goodfellow et al., 2014) pointed out the vulnerability of DNNs to adversarial examples, and proposed an efficient attack, the FGSM, to generate such examples. Since then, as increasingly strong adversaries Carlini & Wagner (2017) are proposed, AT (Madry et al., 2017) is considered as the most effective defense remaining. More recently, Zhang et al. (2019) proposes TRADES, extending AT and showing better robustness than Madry et al. (2017). We refer readers to (Yuan et al., 2019; Biggio & Roli, 2018) for more comprehensive surveys on adversarial attacks and defenses.

However, all AT methods illustrate common drawbacks. Specifically, Rice et al. (2020) shows that AT suffers from severe overfitting that cannot be easily mitigated, which leads to multiple undesirable consequences. For example, AT performs poorly on clean examples, and Song et al. (2018) shows that AT overfits on the adversary it is trained on and performs poorly under other attacks.

**Uncertainty Promotion Techniques in Machine Learning.** The principle of maximum entropy is widely accepted in not only physics, but also machine learning, such as reinforcement learning (Ziebart et al., 2008; Haarnoja et al., 2018). In supervised learning, uncertainty promotion techniques, including entropy maximization (Pereyra et al., 2017) and label smoothing (Szegedy et al., 2016) act on model outputs and can improve generalization in many applications, such as image classification and language modeling (Pereyra et al., 2017; Müller et al., 2019; Szegedy et al., 2016).

There are only several works that study uncertainty promotion techniques in the field of adversarial machine learning. ADP (Pang et al., 2019) claims that diversity promotion in ensembles can promote robustness, for which entropy maximization is used. Essentially, ADP consists of multiple mutually independent models that in total mimic the behavior of entropy maximization. However, ADP only shows improvements under weak attacks, e.g. PGD with 10 iterations. Shafahi et al. (2019) claims that label smoothing can improve robustness beyond AT *by itself*, yet the claim remains to be further investigated under a wider range of adversaries, e.g. adaptive attacks and black-box attacks. As stated by (Uesato et al., 2018; Athalye et al., 2018) that, one should use strong and various attackers for better approximation of the *adversarial risk* (Uesato et al., 2018), it remains unknown how much robustness EntM or LS *alone* can provide under strong and diverse attacks. Moreover, neither of these works study the relationship between AT and uncertainty promotion, which is an important open problem since many regularizers fail to mitigate the overfitting of AT (Rice et al., 2020).

We also discuss knowledge distillation (KD) (Hinton et al., 2015) as another line of work that introduces uncertainty. KD is a procedure that trains a student model to fit the outputs of a teacher network, which are also 'soft' labels and introduce uncertainty. However, KD *alone* has been shown not to improve robustness (Carlini & Wagner, 2016). Also, while (Goldblum et al., 2020) proposed adversarially robust distillation (ARD), combining AT and KD, ARD primarily aims to build robust

models upon robust pre-trained teacher models, while we primarily study uncertainty and do not rely on pre-trained teacher models. Therefore we consider ARD to be orthogonal to our work.

## 3 PRELIMINARIES

**Notations.** We denote a dataset $\mathcal{D} = \{X^{(i)}, y^{(i)}\}_{i=1}^n$, where $X^{(i)} \in \mathbb{R}^d$ are input data, $y^{(i)} \in \mathbb{R}^C$ are one-hot vectors for labels. We denote a DNN parameterized by $\theta$ as $f(X; \theta) : \mathbb{R}^d \to \mathbb{R}^C$, which outputs logits on $C$ classes. We denote $f_\sigma(X; \theta) = \sigma \circ f(X; \theta)$, the network followed by softmax activation. We denote the cross entropy loss as $CE(\hat{y}, y) = -\sum_{i=1}^C y_i \log \hat{y}_i$, the Shannon entropy as $H(p) = -\sum_{i=1}^C p_i \log p_i$ and the Kullback-Leibler divergence as $D_{KL}(p\|q) = \sum_{i=1}^C p_i \log \frac{p_i}{q_i}$.

**Adversarial Attacks and Adversarial Training.** Given a data-label pair $(X, y)$ and a neural network $f(X; \theta)$, adversarial attacks aim to craft an adversarial example $X^{(adv)}, \|X^{(adv)} - X\| \leq \varepsilon$, so as to fool $f(X; \theta)$. Such goal can be formulated as the problem:

$$X^{(adv)} = \arg\max_{X'} L_{atk}(f_\sigma(X'; \theta), y), \text{ subject to} \|X' - X\| \leq \varepsilon. \tag{1}$$

where $L_{atk}$ is some loss function, generally taken as CE. Eqn. 1 can generally be solved via iterative optimization, such as Projected Gradient Descent (PGD) (Madry et al., 2017).

Adversarial Training (AT) is generally considered the most effective defense against adversarial attacks, where the training set consists of both clean examples and adversarial examples, and the following objective is minimized (Goodfellow et al., 2014),

$$\min_\theta \mathbb{E}_{(X,y)\sim\mathcal{D}} \left[ \alpha CE\left(f_\sigma(X; \theta), y\right) + (1-\alpha) \max_{\|X^{(adv)} - X\| \leq \varepsilon} CE\left(f_\sigma(X^{(adv)}; \theta), y\right) \right], \tag{2}$$

where the inner maximization is commonly solved via PGD. We specifically denote AT which uses PGD to solve the inner problem as PAT (Madry et al., 2017). TRADES (Zhang et al., 2019) extends PAT by optimizing Eqn. 3 in similar min-max styles, leading to better robustness than PAT.

$$\min_\theta \mathbb{E}_{(X,y)\sim\mathcal{D}} \left[ CE(f_\sigma(X; \theta), y) + \beta \max_{\|X^{(adv)} - X\| \leq \varepsilon} D_{KL}\left(f_\sigma(X^{(adv)}; \theta)\|f_\sigma(X; \theta)\right) \right]. \tag{3}$$

## 4 METHODOLOGY

### 4.1 ENTROPY MAXIMIZATION

Given an example $(X, y)$ and a neural network $f(X; \theta)$, the proposed entropy maximization (EntM) minimizes the cross entropy loss, while maximizing the Shannon entropy of the output probability.

$$L_{entm}(X, y) = CE\left(f_\sigma(X; \theta), y\right) - \lambda H\left(f_\sigma(X; \theta)\right), \tag{4}$$

where $\lambda$ is a hyperparameter controlling the strength of entropy maximization. For normal training, we minimize over $\mathbb{E}_{(X,y)\sim\mathcal{D}}[L_{entm}(X, y)]$; for AT, such as PAT and TRADES, we replace CE with $L_{entm}$ in Eqn. 2, which for PAT yields Eqn. 5, and similarly for TRADES:

$$\min_\theta \mathbb{E}_{(X,y)\sim\mathcal{D}} \left[ \alpha L_{entm}\left(f_\sigma(X; \theta), y\right) + (1-\alpha) \max_{\|X^{(adv)} - X\| \leq \varepsilon} L_{entm}\left(f_\sigma(X^{(adv)}; \theta), y\right) \right], \tag{5}$$

The entropy maximization term penalizes for over-confident outputs and encourages uncertain predictions, which is different from CE encouraging one-hot outputs. Such a formulation is also seen in (Pereyra et al., 2017; Dubey et al., 2018) in general deep learning and fine-grained classification.

### 4.2 CONNECTION WITH LABEL SMOOTHING

Label smoothing is also a widely used technique that prevents assigning over-confident predictions. Given a data-label pair $(X, y)$, and denote the uniform distribution over $C$ classes as $u_C$, the objective of label smoothing can be formulated as

$$y_{smooth} = y - \gamma(y - u_C), L_{smooth}(X, y) = CE\left((f_\sigma(X; \theta), y_{smooth})\right). \tag{6}$$

Label smoothing and entropy maximization are intrinsically similar. Specifically, both label smoothing and entropy maximization penalize over-confident outputs by:

$$L_{smooth}(X, y) = (1 - \gamma) \operatorname{CE}(f_\sigma(X; \theta), y) + \gamma D_{KL}(u_C \| f_\sigma(X; \theta)) + \gamma \log C. \tag{7}$$

$$L_{entm}(X, y) = \operatorname{CE}(f_\sigma(X; \theta), y) + \lambda D_{KL}(f_\sigma(X; \theta) \| u_C) - \lambda \log C. \tag{8}$$

Therefore EntM and LS are connected, with differences in the asymmetric $D_{KL}(f_\sigma(X; \theta) \| u_C)$ and $D_{KL}(u_C \| f_\sigma(X; \theta))$. In our paper, we carry out experiments on LS and EntM for comparison.

## 5 QUANTITATIVE EXPERIMENTS

### 5.1 EXPERIMENTAL SETTINGS AND RESULTS

**Datasets and Models** We utilize four commonly used datasets for evaluation, CIFAR-10, CIFAR-100, SVHN and MNIST. We train ResNet18, a common selection in adversarial training (Madry et al., 2017) for CIFAR-10, SVHN and CIFAR-100, and a four-layer CNN for MNIST. Details about training settings and architectures can be found in Appendix A.

**Comparison Models** We study the robustness of the following models without adversarial training.

- Normal models trained using standard cross entropy. We denote them as **Normal**.

- **ADP** (Pang et al., 2019), where they built ensembles of models, regularized by maximizing their ensemble diversity. Entropy maximization is involved in the ensemble diversity term. We use 3 models as the ensemble, and take the recommended parameters $(2, 0.5)$.

- Models trained with **EntM** in Eqn. 4. We set $\lambda = 2$ for all datasets following ADP who also set the entropy term to 2.

- Models trained with label smoothing (**LS**). We choose $\gamma$ such that the non-maximal predictions are the same as EntM on each dataset, i.e. $\gamma = 0.74$ for CIFAR-10 and SVHN, and $\gamma = 0.84$ for CIFAR-100.

We also study the robustness of the following models with adversarial training. All adversaries are taken as $L_\infty$ adversary with $\varepsilon = 8/255$, step size $2/255$ with 7 steps for CIFAR-10, CIFAR-100 and SVHN, and $\varepsilon = 48/255$, step size $6/255$ with 10 steps for MNIST.

- AT (Eqn. 2) with PGD adversary, (**PAT**). We take $\alpha = 0.5$ as in (Goodfellow et al., 2014).

- PAT with EntM objective (Eqn. 5), **PAT-EntM**. We set identical $\lambda$ as in EntM..

- PAT with label smoothing (Eqn. 6), **PAT-LS**. We set identical $\gamma$ values as in LS.

- **TRADES** (Zhang et al., 2019), which shows better robustness than PAT. We take $\beta = 3$. [1]

**Threat models.** Without explicit mentioning, we focus on untargeted $L_\infty$ adversaries. We consider PGD adversaries with CE objective (PGD and FGSM), and also CW adversaries (Carlini & Wagner, 2017), i.e. attacks using CW objective and optimized by PGD[2]. We consider both **white** and **black-box** settings. In black-box settings, adversarial examples are crafted on a proxy model trained on identical data and then transferred to the target model (Papernot et al., 2017).

We show test accuracy under various attacks on CIFAR-10, CIFAR-100, SVHN and MNIST in Table 1, 2, 3 and 9[3]. We use FGSM$k$ to denote FGSM with $\varepsilon = k/255$, PGD$p - k$ as $p$-step PGD attacks with $\varepsilon = k/255$, step size $k/2550$, and CW$p - k$ as similar to PGD$p - k$ except for the attack objective. We adopt random initialization with Gaussian $X' = X + N(0, 0.005)$ with 5 restarts for all attacks. For black-box attacks, adversarial examples are crafted on a Normal model for defenses without AT, and on a PAT model for those with AT. Best accuracies higher than random are bolded.

---

[1] We also did experiments on TRADES-EntM, i.e. TRADES trained with entropy maximization objective. The results are shown in Table 12 and 13 in Appendix E.

[2] The C&W adversary is implemented and carried out exactly as (Zhang et al., 2020). Note that ADP builds ensemble models upon probability outputs, CW attacks operating on logits cannot be carried out.

[3] Results on MNIST, and results on PGD20 are shown in Appendix B.

| CIFAR-10 | Attacks | Normal | Without AT | | | With AT | | | |
|---|---|---|---|---|---|---|---|---|---|
| Mode | | | EntM | LS | ADP | PAT | TRADES | PAT-EntM | PAT-LS |
| | Clean | 94.78 | 94.71 | 95.15 | **95.72** | 86.47 | 84.01 | 87.87 | **88.31** |
| | FGSM4 | 37.05 | 72.37 | **74** | 69.5 | 69.21 | 70.33 | **72.96** | **72.95** |
| White-Box | PGD10-4 | 3.27 | *58.45* | ***66.07*** | *42.8* | 68.33 | 68.72 | 71.26 | **71.48** |
| | PGD10-8 | 0 | *37.49* | ***46.12*** | *21.38* | 48.87 | 51.35 | **53.89** | 53.22 |
| | PGD10-16 | 0 | 17.06 | **24.79** | 6.25 | 19.63 | 25.77 | **33.65** | 31.17 |
| | PGD40-4 | 0 | 21.12 | **24.6** | 12.15 | 67.51 | 67.4 | **69.63** | 69.44 |
| | PGD40-8 | 0 | 5.01 | 6.26 | 1.94 | 43.03 | **46.74** | 45.95 | 44.79 |
| | PGD40-16 | 0 | 0.7 | 0.93 | 0.13 | 10.26 | 15.27 | **19.56** | 18.3 |
| | CW40-4 | 0 | 22.76 | **24.8** | - | 67.04 | 66.31 | 68.3 | **68.58** |
| | CW40-8 | 0 | 6.7 | 9.28 | - | 43.63 | **45.78** | 44.52 | 43.75 |
| | CW40-16 | 0 | 1.16 | 2.35 | - | 10.66 | 14.47 | **17.79** | 17.09 |
| Black-Box | PGD10-4 | 27.5 | **35.84** | 32.24 | 31.79 | 76.88 | 75.52 | **79.32** | 79.24 |
| | PGD10-8 | 7.57 | **13.26** | 10.2 | 10.12 | 63.9 | 64.8 | **65.82** | 65.76 |

Table 1: Performance under various attacks on CIFAR-10 (%). 0 denotes $< 0.1\%$. Italics indicate white-box accuracies higher than corresponding black-box ones.

| CIFAR-100 | Attacks | Normal | Without AT | | | With AT | | | |
|---|---|---|---|---|---|---|---|---|---|
| Mode | | | EntM | LS | ADP | PAT | TRADES | PAT-EntM | PAT-LS |
| | Clean | 75.66 | 77.17 | 77.60 | **80.04** | 59.65 | 54.63 | 62.53 | **62.83** |
| | FGSM4 | 12.73 | 28.29 | 29.8 | **31.14** | 38.86 | 37.94 | **44.37** | 43.2 |
| White-Box | PGD10-4 | 0.97 | 9.1 | **9.91** | 6.15 | 37.76 | 37.61 | **43.43** | 42.39 |
| | PGD10-8 | 0 | **3.32** | **3.35** | 1.3 | 22.41 | 24.77 | **29.09** | 27.13 |
| | PGD10-16 | 0 | 0.7 | 1 | 0 | 7.47 | 10.28 | **13.18** | 11.21 |
| | PGD40-4 | 0 | 1.08 | 1.35 | 0.87 | 36.55 | 36.45 | **42.45** | 41.08 |
| | PGD40-8 | 0 | 0.1 | 0 | 0 | 19.62 | 22.6 | **26.1** | 24.05 |
| | PGD40-16 | 0 | 0 | 0 | 0 | 4.73 | 7.26 | **8.22** | 6.69 |
| | CW40-4 | 0 | 0.4 | 0.5 | - | 36.09 | 34.29 | **38.5** | 38.02 |
| | CW40-8 | 0 | 0 | 0 | - | 19.33 | 20.74 | **22.17** | 21.02 |
| | CW40-16 | 0 | 0 | 0 | - | 5.1 | 6.01 | **6.17** | 5.4 |
| Black-Box | PGD10-4 | 13.77 | 18.07 | 18.44 | **21.36** | 49.4 | 46.76 | 52.25 | **52.97** |
| | PGD10-8 | 3.65 | 5.45 | 5.6 | **7.07** | 38.84 | 38.55 | **41.48** | 41.44 |

Table 2: Performance under various attacks on CIFAR-100 (%).

## 5.2 How much do Uncertainty Promotion regularizers improve robustness?

### 5.2.1 Without Adversarial Training

We first analyze the case where no adversarial training is used, namely, EntM, ADP and LS, whose robustness under diverse and strong attacks is under question. We make the following findings.

- Under attacks, EntM and LS perform better than ADP in CIFAR-10 and CIFAR-100, and in SVHN, LS performs better than ADP, while EntM performs similarly as ADP. Therefore we focus our analysis on EntM and LS instead of ADP.

- **EntM and LS show gradient obfuscation under weak attacks.** On CIFAR-10 and SVHN, white-box PGD10 attacks succeed less often than black-box ones (See Table 1 and 3, White- and Black-Box PGD10), which is a clear indication of *gradient obfuscation* (Athalye et al., 2018), i.e. gradient-based attacks perform worse than gradient-free attacks because gradients around a data point are made useless by the model. Gradient obfuscation on EntM and LS leads us to use adversaries with more steps to better approximate their adversarial risk (Uesato et al., 2018), as multi-step adversaries are more likely to escape from

| SVHN Mode | Attacks | Normal | Without AT | | | With AT | | | |
|---|---|---|---|---|---|---|---|---|---|
| | | | EntM | LS | ADP | PAT | TRADES | PAT-EntM | PAT-LS |
| White-Box | Clean | 97.04 | 97.18 | 97.31 | **97.58** | 94 | 93.28 | **94.08** | **94.08** |
| | FGSM4 | 65.74 | 85.39 | **85.75** | 83.55 | 82.99 | **84.26** | 83.21 | 83.04 |
| | PGD10-4 | 27.54 | *72.99* | *79.86* | *70.76* | 79.72 | **80.41** | 80.11 | 79.39 |
| | PGD10-8 | 4.73 | 48.47 | *59.13* | *49.91* | 61.71 | **64.95** | 63.88 | 62.6 |
| | PGD10-16 | 0 | 21.61 | **31.97** | 25.13 | 29.33 | 36.27 | **38.85** | 35.77 |
| | PGD40-4 | 8.37 | 40.85 | **44.53** | 42.73 | 77.71 | 77.96 | **78.3** | 77.06 |
| | PGD40-8 | 0.36 | 10.27 | 12.4 | **13.62** | 52.76 | **57.02** | 54.65 | 52.14 |
| | PGD40-16 | 0 | 1.15 | 1.9 | 2.25 | 14.18 | 17.96 | **23.8** | 17.58 |
| Black-Box | PGD10-4 | 66.07 | **70.47** | 69.76 | 67.83 | 83.69 | 83.78 | 83.85 | **84.08** |
| | PGD10-8 | 43.7 | **49.77** | 48.25 | 46.53 | 68.88 | 69.4 | 69.03 | **69.48** |

Table 3: Performance under various attacks on SVHN (%). Italics indicate white-box accuracies higher than corresponding black-box ones.

the area where gradient obfuscation happens and find adversarial examples. We provide more evidence of gradient obfuscation in Appendix C.

- **EntM and LS improve robustness only under small perturbations.** We carry out evaluations on PGD40. It can be shown that improvements over random guessing can only be seen under PGD40-4 on CIFAR-10 and SVHN (Table 1 and 3). Therefore, the claims by (Pang et al., 2019; Shafahi et al., 2019) that ADP and LS improve robustness are conditional, and EntM, LS and ADP are only robust under small perturbations.

### 5.2.2 WITH ADVERSARIAL TRAINING

We study AT regularized by uncertainty promotion regularizers and make the following findings.

- PAT-EntM and PAT-LS outperform PAT on clean examples by 1% on CIFAR-10, 3% on CIFAR-100, and under strong attacks like PGD40, outperform PAT by 2-9% on CIFAR-10, and 3-6% on CIFAR-100. See Clean and PGD40 in Table 1, 3, 2.

- PAT-EntM achieves generally similar or better performance than its counterpart PAT-LS, e.g. on CIFAR-10 and CIFAR-100, PAT-EntM achieves 1% better adversarial accuracy, while PAT-LS achieves <1% better accuracy on clean examples, see Table 1 and 2. Therefore for the rest of the paper we primarily focus on PAT-EntM.

- Robustness improvements are most evident under large perturbations. For example, accuracies under PGD40-16 improved by 9% on CIFAR-10 and SVHN from PAT to PAT-EntM, see Table 1 and 3, which is exactly the opposite from where AT is not used.

- PAT-EntM is generally more robust than TRADES by over 3% on CIFAR-10 and CIFAR-100, and is similarly robust as TRADES on SVHN, further showing the effectiveness of uncertainty promotion regularizers.

- Results under CW (Table 1 and 2) show that PAT-EntM is still more robust than PAT (by 7% on CIFAR-10 and 3% on CIFAR-100), and TRADES (except CIFAR10, PGD40-8). The results under a different adversary further verifies the robustness improvement.

In addition, we train WideResNet34-10 (Zagoruyko & Komodakis, 2016) following TRADES, and evaluate the robustness of PAT, TRADES and PAT-EntM. We show the results in Appendix D, Table 11.

Following (Athalye et al., 2018), we perform the following sanity checks to make sure PAT-EntM is not causing gradient obfuscation, and to better approximate and evaluate its adversarial risk.

**Perturbation-Accuracy Curve.** We expand perturbations to test whether the robust accuracy monotonically decreases, and whether sufficiently large perturbations lead to 0 accuracy. We utilize PGD with $\varepsilon = k/255$, step size $2/255$ and $k$ iterations, $k \in [10, 150]$, on PAT, PAT-EntM and TRADES, on CIFAR-10 and CIFAR-100. We show the results in Fig. 1. On both datasets and under

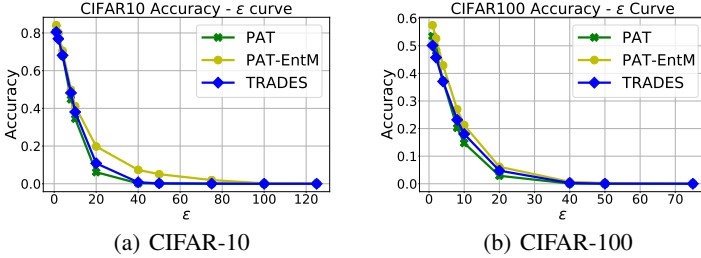

Figure 1: Accuracy-$\varepsilon$ curve on CIFAR-10 and CIFAR-100. x-axis values denote pixel changes.

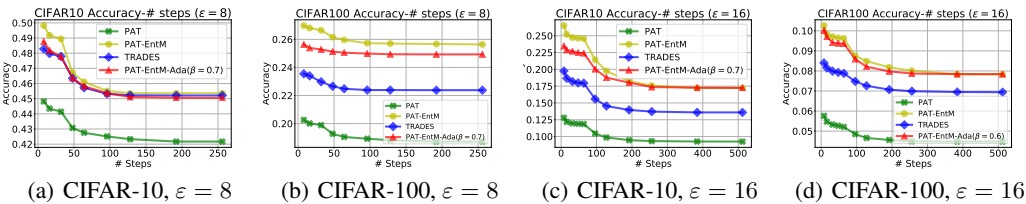

Figure 2: Accuracy-# step curve on CIFAR-10 and CIFAR-100. $\varepsilon = 8/255, 16/255$

all $\varepsilon$, PAT-EntM outperforms PAT and TRADES. Moreover, under sufficiently large $\varepsilon$, all accuracies reach 0, which is another evidence that PAT-EntM is not obfuscating gradients.

**Attacks with more iterations.** We perform attacks with fixed $\varepsilon$ and more iterations to make sure attacks converge, and to reliably evaluate the robustness of PAT-EntM. We take $\varepsilon = 8/255, 16/255$, with # steps $k \in [8, 256]$, step size $\max(1/510, 2\varepsilon/k)$ on CIFAR-10 and CIFAR-100.

We show the results in Fig. 2. It can be shown that even when the attacks converge at 200 steps, PAT-EntM still consistently outperform TRADES (over 4% on both CIFAR-10 and CIFAR-100). Moreover, PAT-EntM is consistently similarly robust or more robust than TRADES.

**Adaptive Attacks.** We perform adaptive attacks on PAT-EntM. Due to its connection with label smoothing, we leverage CE with smoothed labels in Eqn. 6 as the loss function $L_{atk}$ in Eqn. 1. We search for the best parameter $\gamma$ for each attack from $[0, 1]$ at an interval of 0.1. We consider $\varepsilon = 8/255, 16/255$ on CIFAR-10 and CIFAR-100, and plot corresponding curves also in Fig. 2.

With properly selected smooth parameter $\gamma$, adaptive attacks can succeed more often than non-adaptive ones. However, even under strong adaptive attacks, we can still see improvements of PAT-EntM over PAT (7% on CIFAR-10 and 4% on CIFAR-100, $\varepsilon = 16/255$) and in most cases over TRADES, except for CIFAR-10, $\varepsilon = 8/255$, where robust accuracies of PAT-EntM and TRADES are 45.05% and 45.21%, respectively.

### 5.3 HYPERPARAMETER ANALYSIS

We vary the hyperparameter $\lambda$ in Eqn. 4 and 5 to see how the uncertainty level $\lambda$ influences accuraries on both clean and adversarial examples. We vary $\lambda \in [0, 10]$ on CIFAR-10 and CIFAR-100.

We plot the accuracies on clean examples and under PGD40-8 in Fig. 3. The results show that on both datasets, accuracies on both clean and adversarial examples simultaneously increase with $\lambda$ from 0.1 to 5. This is an appealing property and contrary to the belief that accuracy and robustness are at odds with each other (Zhang et al., 2019; Tsipras et al., 2018). We also observe in Fig. 3 that on both datasets, $\lambda = 2$ and $\lambda = 5$ achieve similar performances (within 0.5%) in terms of both accuracy and robustness, which shows that EntM is not highly sensitive to hyperparameters.

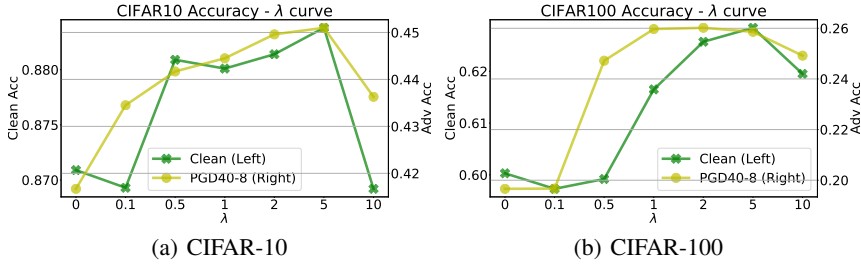

Figure 3: Hyperparameter Analysis for $\lambda$.

## 6 HOW UNCERTAINTY PROMOTION WORKS - FURTHER ANALYSIS

In this section we dig deeper into how uncertainty promotion works, which may bring deeper insights towards more robust algorithms. For brevity, we use $f(X)$ as a shorthand for $f(X; \theta)$ and omit $\theta$.

We define $M_{f,X} = f(X)_y - \max_{i \neq y} f(X)_i$ as the decision margin of $f$ at point $X$, and introduce Theorem 1 connecting robustness with $M_{f,X}$ and the Lipschitz constant of $f$.

**Theorem 1** (Tsuzuku et al. (2018)). *Denote the global Lipschitz constant of $f(X; \theta)$ as $l_f$. The following holds for any data point $X$, and any perturbation $\delta$.*

$$M_{f,X} \geq \sqrt{2}\|\delta\|_2 l_f \Rightarrow M_{f,X+\delta} \geq 0. \tag{9}$$

Theorem 1 shows that, for an arbitrary noise $\delta$, as long as $\|\delta\|_2 \leq \frac{M_{f,X}}{\sqrt{2}l_f}$, the model $f$ will output correct predictions on $X + \delta$. Therefore, we focus on the metric *normalized margin*, defined as $\frac{M_{f,X}}{l_f}$ for analyzing robustness. However, as $l_f$ is hard to compute, we then introduce an approximation for the analysis. We focus on the first-order approximation of $f(X)$,

$$f(X') \approx f(X) + \nabla_X f(X)^T (X' - X) \tag{10}$$

If the approximation holds well locally, we have the following inequality:

$$\|f(X') - f(X)\|_2 \approx \|\nabla_X f(X)^T (X' - X)\|_2 \leq \|\nabla_X f(X)\|_2 \|X' - X\|_2, \tag{11}$$

$$\|\nabla_X f(X)\|_2 \geq \frac{\|f(X) - f(X')\|_2}{\|X' - X\|_2} \tag{12}$$

where $\|\nabla_X f(X)\|_2$ is the spectral norm (also the largest singular value) of the Jacobian matrix. Eqn. 12 shows that $\|\nabla_X f(X)\|_2$ *locally* upper bounds the local Lipschitz constant of $f$ at point $X$ (the R.H.S of Eqn.12). Therefore, we compute $\frac{M_{f,X}}{\|\nabla_X f(X)\|_2}$ via differentiation to analyze robustness.

However, since $f(X)$ is non-linear, we define another term $Q_f(X', X)$ describing how much $f(X)$ deviates from the linear approximation, to account for cases where Eqn.10 approximates poorly.

$$Q_f(X', X) = \frac{\|f(X') - f(X) - \nabla_X f(X)^T (X' - X)\|_2}{\|\nabla_X f(X)^T (X' - X)\|_2}, \|X' - X\|_2 \leq \varepsilon. \tag{13}$$

We sample 2,000 test samples from CIFAR-10, and study the relationship among *normalized margins*, *non-linearity* $Q_f(X', X)$, and robustness. We use $L_2$ adversaries in correspondence with Theorem 1. We list related results in Table 4. With EntM, $\|\nabla_X f(X)\|_2$ is shrunken by over 10 times regardless whether AT is used, leading to a 4-time increase of normalized margin in both EntM and PAT-EntM. However, while EntM alone achieves high regularized margin, it also massively compromises local linearity compared to Normal, as shown by the high $Q_f(X^{(adv)}, X)$. Therefore, the area where the linear approximation in Eqn. 10-12 holds is limited, thereby unable to guarantee robustness. By contrast, PAT-EntM only worsens local linearity slightly compared to PAT, and therefore the enlarged normalized margin contributes to better robustness.

| Models | Normal | EntM | PAT | PAT-EntM |
|---|---|---|---|---|
| $M_{f,X}$ | 10.55 | 1.3054 | 6.12 | 1.12 |
| $\|\nabla_X f(X)\|_2$ | 72.26 | 3.44 | 10.69 | 0.84 |
| $\frac{M_{f,X}}{\|\nabla_X f(X)\|_2}$ | 0.1919 | 2.8132 | 0.6398 | 2.9913 |
| Accuracy (PGD20-0.5) | 0.0029 | 0.166 | 0.5909 | 0.6142 |
| $Q_f(X^{(adv)}, X)$ | 4.4 | 52.51 | 4.59 | 5.36 |
| Correlation$\left(\frac{M_{f,X}}{\|\nabla_X f(X)\|_2}, \|X^{(adv)} - X\|_2\right)$ | 0.8327 | 0.4794 | 0.8216 | 0.8131 |

Table 4: Analysis of margin $M_{f,X}$, Jacobian matrix $\nabla_X f(X; \theta)$ and non-linearity $Q_f(X^{(adv)}, X)$.

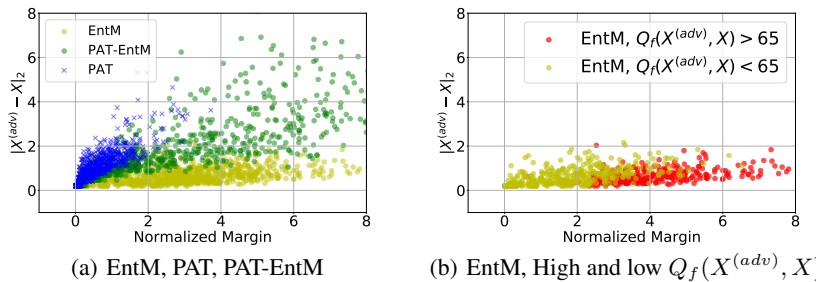

(a) EntM, PAT, PAT-EntM  (b) EntM, High and low $Q_f(X^{(adv)}, X)$

Figure 4: Relationship between normalized margin and distortion for EntM, PAT and PAT-EntM.

We further show that normalized margin, *along with local linearity* explains better robustness. We leverage an infinite $L_2$ attack on each example $X$ until misclassification. We then compute the distortions $\|X^{(adv)} - X\|_2$, and study the relationship between distortions and normalized margins. A more robust model should require a larger distortion to successfully attack.

We plot the normalized margins and distortions in Fig 4(a), and show their Pearson correlations in Table 4. On Normal, PAT, PAT-EntM, the correlation is much higher than that on EntM. Also, Fig. 4(a) shows that although EntM has similar normalized margins as PAT-EntM, the distortions required are smaller on EntM than on PAT-EntM. Since linearity is not only the key in the derivations of Eqn. 12, but also a major difference of PAT-EntM from EntM, these results indicate that high non-linearity compromises robustness on EntM.

We further separate points with $Q_f(X^{(adv)}, X) > 65$ on EntM, which is the 40-percentile, and plot them separately in Fig. 4(b). Samples with higher local non-linearity have higher normalized margins as PAT-EntM, but no higher distortions and hence no better robustness. All results show that only when combined with local linearity, normalized margins explains robustness.

## 7 CONCLUSION

In this paper, we revisit uncertainty promotion regularizers in the field of adversarial learning. We find out that uncertainty promotion regularizers alone are causing gradient obfuscation, and that they alone only provide inconsistent robustness against small perturbations. Contrarily, our extensive experiments demonstrate that uncertainty promotion regularizers augment AT, improving accuracy on clean examples and enhancing robustness, especially under large perturbations. We further demonstrate that both good local linearity and shrunken norm of Jacobian matrices contribute to better robustness shown by PAT-EntM than PAT.

We hope that this paper would again underscore the necessity to evaluate under strong attacks, and raise attention to further insights about why uncertainty promotion regularizers work. We consider theoretical investigations of why they work well alongside AT as an important future work.

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

## A  DETAILED EXPERIMENTAL SETTINGS

### A.1  ARCHITECTURES

We train ResNet18 for CIFAR-10, CIFAR-100 and SVHN. We use the same architecture of ResNet18 as in TRADES[4]. The input size for ResNet18 is $32 \times 32 \times 3$.

We train a four-layer CNN for MNIST. The detailed architecture is:

- Input: $28 \times 28 \times 1$ grayscale images.
- 5x5 convolution with stride 5 and 10 filters. ReLU activation.
- Max pooling with stride 2.
- 5x5 convolution with stride 5 and 20 filters. ReLU activation.

---

[4]https://github.com/yaodongyu/TRADES/blob/master/models/resnet.py

- Max pooling with stride 2.
- Flatten.
- FC layer with input shape 320 and output 10. ReLU activation.
- FC layer with input shape 50 and output shape 10.

## A.2 TRAINING PARAMETER SETTINGS

| Settings/Models | Four-layer | ResNet18, without AT | ResNet18, with AT |
|---|---|---|---|
| Input Scale | | $[0, 1]$ | |
| Input Size | $28 \times 28 \times 1$ | $32 \times 32 \times 3$ | $32 \times 32 \times 3$ |
| Total Training Epochs | 40 | 200 | 120 |
| Weight Decay | $10^{-4}$ | $10^{-4}$ | $10^{-4}$ |
| Optimizer | | SGD with momentum 0.9 | |
| Initial Learning Rate | 0.01 | 0.1 | 0.1 |
| Learning Rate Decay Rate | 0.1 | 0.1 | 0.1 |
| Learning Rate Decay Epoch | 20 | $[100, 150]$ | $[60, 84, 100]$ |
| Early Stopping | No | No | No |
| Batch Size | 128 | 64 | 64 |

Table 5: Training Settings

We list all training settings in Table 5. For each quantitative result, we train 3 models, and each PGD is repeated 3 times for average.

For datasets, we perform standard data augmentation techniques, including random cropping and random horizontal flipping (on CIFAR-10 and CIFAR-100).

## B FULL EXPERIMENTAL RESULTS

| CIFAR-10 | Attacks | Normal | Without AT | | | With AT | | | |
|---|---|---|---|---|---|---|---|---|---|
| Mode | | | EntM | ADP | LS | PAT | PAT-EntM | PAT-LS | TRADES |
| | Clean | 94.78 | 94.71 | **95.72** | 95.15 | 86.47 | 87.87 | **88.31** | 84.01 |
| | FGSM4 | 37.05 | 72.37 | 69.5 | **74** | 70.33 | **72.96** | **72.95** | 69.21 |
| White-Box | PGD10-4 | 3.27 | *58.45* | *42.8* | ***66.07*** | 68.33 | 71.26 | **71.48** | 68.72 |
| | PGD10-8 | 0 | *37.49* | *21.38* | ***46.17*** | 48.87 | **53.89** | 53.22 | 51.35 |
| | PGD10-16 | 0 | 17.06 | 6.25 | **24.79** | 19.63 | **33.65** | 31.17 | 25.77 |
| | PGD20-4 | 0.31 | *36.73* | 22.38 | ***47.86*** | 67.89 | **70.1** | 69.88 | 67.58 |
| | PGD20-8 | 0 | *14.59* | *5.8* | ***23.11*** | 44.61 | **48.72** | 47.4 | 47.82 |
| | PGD20-16 | 0 | 3.86 | 0.82 | 7.95 | 12.76 | **24.79** | 22.73 | 18.27 |
| | PGD40-4 | 0 | 21.12 | 12.15 | **24.6** | 67.51 | **69.63** | 69.44 | 67.4 |
| | PGD40-8 | 0 | 5.01 | 1.94 | 6.26 | 43.03 | 45.95 | 44.79 | **46.74** |
| | PGD40-16 | 0 | 0.7 | 0.13 | 0.93 | 10.26 | **19.56** | 18.3 | 15.27 |
| Black-Box | PGD10-4 | 27.5 | **35.84** | 31.79 | 32.24 | 76.88 | **79.32** | 79.24 | 75.52 |
| | PGD10-8 | 7.57 | **13.26** | 10.12 | 10.2 | 63.9 | **65.82** | 65.76 | 64.8 |
| | PGD20-4 | 19.73 | **28.53** | 24.78 | 26.75 | 76.73 | **78.67** | **78.69** | 74.84 |
| | PGD20-8 | 3.12 | **7.43** | 5.36 | 6.46 | 62.24 | **64.44** | 64 | 63.65 |

Table 6: Performance under various attacks on CIFAR-10 (%). 0 denotes $< 0.1\%$. Italics indicate white-box accuracies higher than corresponding black-box ones.

We show full experimental results in 6, 7, 8 and 9. Although in MNIST, the improvement is less significant (1% or below), and TRADES significantly outperforms both PAT and PAT-EntM, PAT-EntM still achieves improvement over AT in terms of robustness under various attacks.

| CIFAR-100 Mode | Attacks | Normal | Without AT | | | With AT | | | |
|---|---|---|---|---|---|---|---|---|---|
| | | | EntM | ADP | LS | PAT | PAT-EntM | PAT-LS | TRADES |
| | Clean | 75.66 | 77.17 | **80.04** | 77.60 | 59.65 | 62.53 | **62.83** | 54.63 |
| | FGSM4 | 12.73 | 29.8 | 28.29 | **31.14** | 38.86 | **44.37** | 43.2 | 37.94 |
| White-Box | PGD10-4 | 0.97 | 9.1 | 6.15 | **9.91** | 37.76 | **43.43** | 42.39 | 37.61 |
| | PGD10-8 | 0 | **3.32** | 1.3 | **3.35** | 22.41 | **29.09** | 27.13 | 24.77 |
| | PGD10-16 | 0 | 0.7 | 0 | 1 | 7.47 | **13.18** | 11.21 | 10.28 |
| | PGD20-4 | 0 | 2.9 | 1.94 | **3.15** | 36.89 | **42.65** | 41.3 | 36.83 |
| | PGD20-8 | 0 | 0.69 | 0.15 | 0.87 | 20.46 | **26.89** | 24.75 | 23.12 |
| | PGD20-16 | 0 | 0 | 0 | 0 | 5.43 | **9.82** | 8.02 | 8.01 |
| | PGD40-4 | 0 | 1.08 | 0.87 | 1.35 | 36.55 | **42.45** | 41.08 | 36.45 |
| | PGD40-8 | 0 | 0.1 | 0 | 0 | 19.62 | **26.1** | 24.05 | 22.6 |
| | PGD40-16 | 0 | 0 | 0 | 0 | 4.73 | **8.22** | 6.69 | 7.26 |
| Black-Box | PGD10-4 | 13.77 | 18.07 | 21.36 | 18.44 | 49.4 | 52.25 | **52.97** | 46.76 |
| | PGD10-8 | 3.65 | 5.45 | **7.07** | 5.6 | 38.84 | **41.48** | 41.44 | 38.55 |
| | PGD20-4 | 9.92 | 13.31 | **16.12** | 13.5 | 49.34 | 52.02 | **52.56** | 46.45 |
| | PGD20-8 | 1.89 | 2.84 | **3.7** | 2.98 | 38.25 | **40.66** | 40.39 | 38.13 |

Table 7: Performance under various attacks on CIFAR-100 (%). 0 denotes $< 0.1\%$

| SVHN Mode | Attacks | Normal | Without AT | | | With AT | | | |
|---|---|---|---|---|---|---|---|---|---|
| | | | EntM | ADP | LS | PAT | PAT-EntM | PAT-LS | TRADES |
| | Clean | 97.04 | 97.18 | **97.58** | 97.31 | 94 | **94.08** | **94.08** | 93.28 |
| | FGSM4 | 65.74 | 85.39 | 83.55 | **85.75** | 82.99 | 83.21 | 83.04 | **84.26** |
| White-Box | PGD10-4 | 27.54 | *72.99* | *70.76* | **79.86** | 79.72 | 80.11 | 79.39 | **80.41** |
| | PGD10-8 | 4.73 | 48.47 | *49.91* | **59.13** | 61.71 | 63.88 | 62.6 | **64.95** |
| | PGD10-16 | 0 | 21.61 | 25.13 | **31.97** | 29.33 | **38.85** | 35.77 | 36.27 |
| | PGD20-4 | 13.04 | 55.47 | 54.68 | **61.34** | 78.15 | **78.66** | 77.55 | 78.45 |
| | PGD20-8 | 0.81 | 22.42 | 25.24 | **27.35** | 55.24 | 57.37 | 55.56 | **58.13** |
| | PGD20-16 | 0 | 5.0 | 6.6 | 7.2 | 18.65 | **27.57** | 23.9 | 24.49 |
| | PGD40-4 | 8.37 | 40.85 | 42.73 | **44.53** | 77.71 | **78.3** | 77.06 | 77.96 |
| | PGD40-8 | 0.36 | 10.27 | **13.62** | 12.4 | 52.76 | 54.65 | 52.14 | **57.02** |
| | PGD40-16 | 0 | 1.15 | 2.25 | 1.9 | 14.18 | **23.8** | 17.58 | 17.96 |
| Black-Box | PGD10-4 | 66.07 | **70.47** | 67.83 | 69.76 | 83.69 | 83.85 | **84.08** | 83.78 |
| | PGD10-8 | 43.7 | **49.77** | 46.53 | 48.25 | 68.88 | 69.03 | **69.48** | 69.4 |
| | PGD20-4 | 60.17 | **66.33** | 63.79 | 65.8 | 82.67 | 83.15 | **83.3** | 82.94 |
| | PGD20-8 | 36.1 | **45.3** | 40.18 | 43.05 | 65.00 | 65.42 | **65.43** | 65.32 |

Table 8: Performance under various attacks on SVHN (%). 0 denotes $< 0.1\%$. Italics indicate white-box accuracies higher than corresponding black-box ones.

## C    GRADIENT OBFUSCATION OF LS AND ENTM WITHOUT AT

In this section we present more evidence regarding gradient obfuscation of EntM and LS.

### C.1    RANDOM SEARCHING

We present other evidence that EntM, LS and ADP alone may sometimes lead to gradient obfuscation. We sample 1000 images from the test set of CIFAR-10. We carry out random search attacks for 10,000 times on each image where EntM, LS and ADP succeeded in defending, and list the success rate of random attacks in Table 10. In many seemingly successful defenses (15% on LS, 6% on ADP and EntM), gradient-based attackers are obfuscated that they cannot find adversarial examples while such examples do exist.

| MNIST | Attacks | Normal | Without AT | | With AT | | |
|---|---|---|---|---|---|---|---|
| Mode | | | EntM | ADP | PAT | PAT-EntM | TRADES |
| White-Box | Clean | 99.2 | 99.13 | **99.37** | **98.47** | 98.04 | 97.78 |
| | FGSM32 | 55.31 | 62.77 | **77.61** | 95.14 | 95.01 | **95.18** |
| | PGD10-32 | 26.03 | 32.73 | **57.54** | 94.15 | 94.1 | **94.28** |
| | PGD10-48 | 2.35 | 3.08 | **17.06** | 89.82 | **90.65** | 90.57 |
| | PGD20-32 | 20.38 | 23.32 | **43.76** | 93.91 | 93.91 | **94.2** |
| | PGD20-48 | 1.13 | 1.28 | 7.34 | 88.62 | 89.77 | **90.53** |
| | PGD40-32 | 19.04 | 19.89 | **39.14** | 93.54 | 93.88 | **94.17** |
| | PGD40-48 | 0.98 | 0.85 | 4.52 | 88.39 | 89.59 | **90.52** |
| Black-Box | PGD10-32 | 83.11 | 85.76 | **87.07** | 96.32 | 96.16 | 96.93 |
| | PGD10-48 | 46.94 | 52.2 | **52.77** | 94.6 | 94.48 | **95.79** |
| | PGD20-32 | 81.43 | 83.95 | **84.88** | 96.31 | 95.96 | **96.81** |
| | PGD20-48 | 44.68 | 46.23 | **46.81** | 94.36 | 94.23 | **95.45** |

Table 9: Performance under various attacks on MNIST (%).

| Models | EntM | ADP | LS | PAT |
|---|---|---|---|---|
| # PGD8 fails | 368 | 289 | 487 | 493 |
| Among which # Random succeeds | 20 | 12 | 70 | 0 |

Table 10: Attack success rate of random searching attacks on CIFAR-10.

We also perform exactly the same experiments for PAT and PAT-EntM. For samples where PAT and PAT-EntM succeeded in defending, *none* of them can be successfully attacked via a 10,000-time random search attack. It endorses that PAT-EntM is not suffering from gradient obfuscation.

## C.2 Loss Surface Visualization

We choose data-label pairs $(X, y)$ from the CIFAR-10 test set, and plot its local loss surface at $X' = X + \epsilon_1 d_1 + \epsilon_2 d_2$ to visually show the properties of EntM and PAT-EntM. $d_1 = \text{sign}(\nabla_X \text{CE}(f_\sigma(X; \theta), y))$ is the gradient sign direction, and $d_2$ is a random direction with $\|d_2\|_\infty = 1$. We choose $\epsilon_1, \epsilon_2 \in [-0.04, 0.04]$.

Among the first 8 samples in the CIFAR-10 test set, we select 2 of them and plot them in Fig. 5. It can be shown that EntM alone is likely (25% would be a rough estimation. ) to create a 'hole' in a small neighborhood of $X$, thereby causing gradient-based iterative attackers to stuck. This result corresponds with the fact that weak gradient-based attackers cannot achieve a high success rate due to gradient obfuscation. It also corresponds to the results in Table 4 that EntM alone may compromise local linearity by creating a highly curved surface.

We also plot the decision margin $M_{f,X'}$ at $X' = X + \epsilon_1 d_1 + \epsilon_2 d_2$ in Fig. 6. It can be shown that although EntM alone creates a small neighborhood where the margin is large, the area of the neighborhood is small, leaving large areas where $M_{f,X} = 0$. It shows that EntM alone cannot ensure robustness, especially when perturbations are large, and corresponds with results in Table 1, 2, 3.

## D Results on WideResNet34-10

Following TRADES (Zhang et al., 2019), we train WideResNet34-10 (Zagoruyko & Komodakis, 2016) with PAT, PAT-EntM and TRADES on CIFAR-10 and CIFAR-100 using the same procedure as Appendix A. We use $\lambda = 1$ for EntM on CIFAR-10, and $\lambda = 0.5$ on CIFAR-100, which is chosen from $[0.5, 1, 2, 5]$. We report accuracy and robustness results in Table 11.

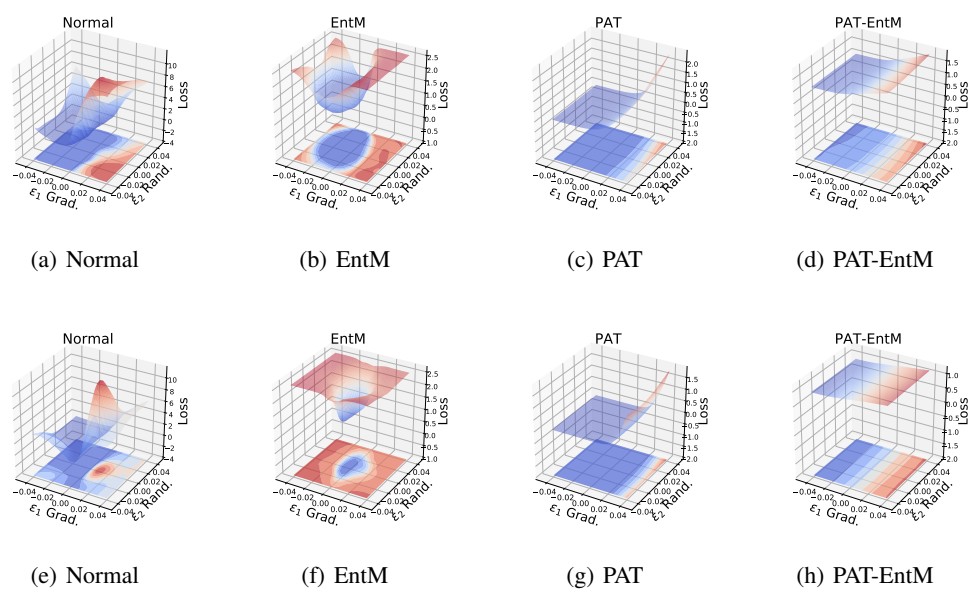

Figure 5: Loss surface visualization at $X' = X + \epsilon_1 d_1 + \epsilon_2 d_2$ for for two $X$ taken from the first 8 images in CIFAR-10 test set.

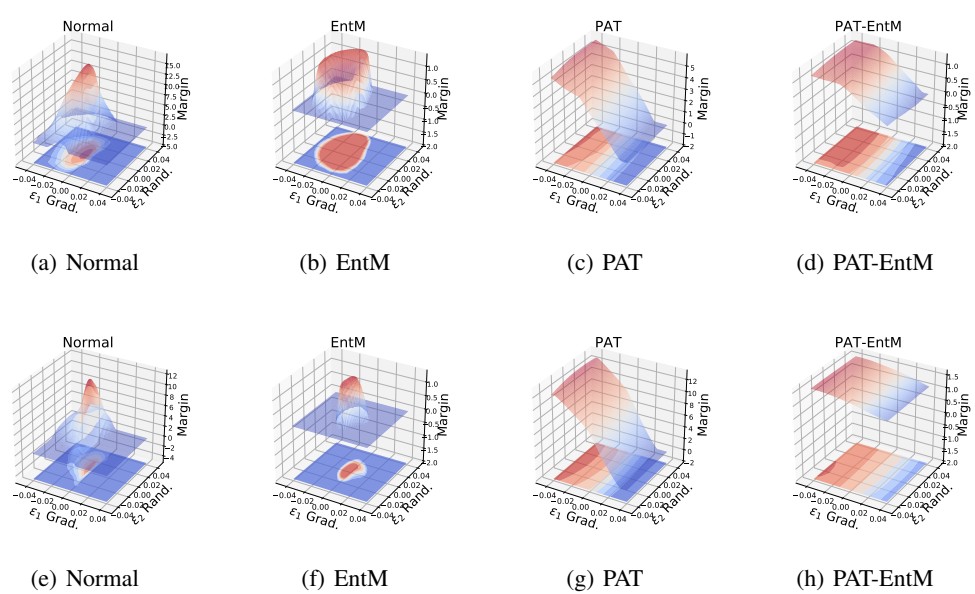

Figure 6: Decision margin visualization at $X' = X + \epsilon_1 d_1 + \epsilon_2 d_2$ for two $X$ taken from the first 8 images in CIFAR-10 test set.

| | CIFAR-10 | | | CIFAR-100 | | |
|---|---|---|---|---|---|---|
| | PAT | TRADES | PAT-EntM | PAT | TRADES | PAT-EntM |
| Clean | 88.11 | 86.09 | **88.65** | 61.96 | 58.57 | **63.93** |
| PGD40-4 | 69.2 | 68.09 | **69.98** | 39.49 | 38.52 | **42.85** |
| PGD40-8 | 45.53 | 46.7 | **52.98** | 22.89 | 23.49 | **26.43** |
| PGD40-16 | 19.03 | 19.27 | **33.21** | 6.2 | 7.64 | **10.6** |
| CW40-4 | 69.42 | 68.33 | **69.66** | 39.44 | 38.2 | **40.86** |
| CW40-8 | 46.23 | 47.76 | **50.4** | 23.47 | 23.78 | **24.7** |
| CW40-16 | 18.9 | 18.09 | **31.31** | 6.64 | **7.93** | 7.77 |

Table 11: Results on WideResNet34-10 of PAT, PAT-EntM and TRADES

| CIFAR-10 | | PAT | TRADES | PAT-EntM | TRADES-EntM |
|---|---|---|---|---|---|
| | Clean | 86.47 | 84.01 | **87.87** | 85.76 |
| | FGSM4 | 70.33 | 69.21 | **72.96** | 70.43 |
| White-Box | PGD10-4 | 68.33 | 68.72 | **71.48** | 69.48 |
| | PGD10-8 | 48.87 | 51.35 | **53.89** | 51.56 |
| | PGD10-16 | 19.63 | 25.77 | **33.65** | 26.47 |
| | PGD20-4 | 67.89 | 67.58 | **70.1** | 68.83 |
| | PGD20-8 | 44.61 | 47.82 | **48.72** | 48.18 |
| | PGD20-16 | 12.76 | 18.27 | **24.79** | 19.01 |
| | PGD40-4 | 67.51 | 67.4 | **69.63** | 68.64 |
| | PGD40-8 | 43.03 | **46.74** | 45.95 | 46.43 |
| | PGD40-16 | 10.26 | 15.27 | **19.56** | 15.14 |
| Black-Box | PGD10-4 | 76.88 | 75.52 | **79.32** | 77.09 |
| | PGD10-8 | 63.9 | 64.8 | **65.82** | 65.10 |
| | PGD20-4 | 76.73 | 74.84 | **78.67** | 76.81 |
| | PGD20-8 | 62.24 | 63.65 | **64.44** | 63.92 |

Table 12: TRADES-EntM in comparison with PAT, TRADES and PAT-EntM on CIFAR-10

We find out that by using large networks, the amount of uncertainty needed decreases. On the optimal level of uncertainty $\lambda = 1$ and $0.5$ respectively, we can still observe consistent improvements with respect to both accuracies on clean examples and under attacks.

# E TRADES WITH ENTM

We also carry out experiments on CIFAR-10 and CIFAR-100 to see how EntM works with TRADES. We take $\lambda = 2$ and list the results in Table 12 and 13. It can be shown that TRADES-EntM improves accuracy on clean examples by 6% on CIFAR-100 and 1.7% on CIFAR-10 compared to TRADES, but the robustness improvements are less evident than PAT-EntM, e.g. less than 1% improvement compared to TRADES under attacks on CIFAR-10, and 2-4% improvements on CIFAR-100.

| CIFAR-100 | | PAT | TRADES | PAT-EntM | TRADES-EntM |
|---|---|---|---|---|---|
| | Clean | 59.65 | 54.63 | **62.53** | 60.96 |
| | FGSM4 | 38.86 | 37.94 | **44.37** | 42.84 |
| | PGD10-4 | 37.76 | 37.61 | **42.39** | 41.96 |
| | PGD10-8 | 22.41 | 24.77 | **29.09** | 27.78 |
| White-Box | PGD10-16 | 7.47 | 10.28 | **13.18** | 12.33 |
| | PGD20-4 | 36.89 | 36.83 | **42.65** | 41.4 |
| | PGD20-8 | 20.46 | 23.12 | **26.89** | 25.99 |
| | PGD20-16 | 5.43 | 7.26 | **9.82** | 9.16 |
| | PGD40-4 | 36.55 | 36.45 | **42.45** | 41.22 |
| | PGD40-8 | 19.62 | 22.6 | **26.1** | 25,34 |
| | PGD40-16 | 4.73 | 7.26 | **8.22** | 7.9 |
| | PGD10-4 | 49.4 | 46.76 | **52.25** | 50.08 |
| | PGD10-8 | 38.84 | 38.55 | **41.48** | 39.37 |
| Black-Box | PGD20-4 | 49.34 | 46.45 | **52.02** | 48.12 |
| | PGD20-8 | 38.25 | 38.13 | **40.66** | 39.01 |

Table 13: TRADES-EntM in comparison with PAT, TRADES and PAT-EntM on CIFAR-100

