# OpenReview forum: "Rethinking Uncertainty in Deep Learning: Whether and How it Improves Robustness"
_ICLR.cc/2021/Conference — Reject_

### Official Review · AnonReviewer4 · 2020-10-15
**Limited novelty and insufficient experiments.**

**Rating:** 4
**Confidence:** 5

**Review:**

1) First, this work studies how entropy maximization and label smoothing combined with adversarial training can improve adversarial robustness. Although these two techniques have been shown to prevent model from being over-confident, I still think it is an over-claim that "rethinking uncertainty in deep learning" as this work does not truly study uncertainty in deep learning or the correlation between uncertainty and adversarial robustness.
2) Label smoothing alone does not provide stronger adversarial robustness, this is not a surprising result as pointed out by many existing work. As entropy maximization is similar as label smoothing, it is also in the expectation that they have similar performance without providing stronger adversarial robustness.
3) "Combining label smoothing/EntM with adversarial training can further provide stronger adversarial robustness", this is the main conclusion of this work. However, I am not fully convinced this is novel enough as this has been observed by existing work. In addition, this work only tested the model on the PGD attacks, the same type of attack (although with smaller l infinity norm bound) is used during training. It is very necessary to test the model against different types of attacks, especially decision-boundary based attacks, to support this conclusion as this is the main contribution of this work.
In all, I vote for a rejection for this work.

******** After Rebuttal ************
I carefully read the authors' response and unfortunately they do not address my concerns. Based on my research background in adversarial robustness and uncertainty estimates, I would keep my original rating unchanged as this work has very limited contribution to these two areas.

---

> ### Author Response · Authors · 2020-11-13
> **Thanks for your comments. We provide our evidence and look forward to additional clarifications from your side.**
>
> We are grateful to you for your suggestions on this paper, and we provide some of our evidence in support of our claims. We are also looking forward to some additional clarifications from your side.
>
> 1. Regarding your opinion on **over-claiming**.
> You point out that "this work does not **truly** study uncertainty in deep learning or the correlation between uncertainty and adversarial robustness". We would like to point out that entropy, "play a central role in information theory as measures of information, choice and **uncertainty**" (Shannon, 1948), and therefore, from our perspective, studying entropy in supervised learning should be considered as studying "uncertainty".
>
> Also, through our experiments, we find out that adding uncertainty to adversarial training is helpful in improving both accuracy and robustness. In addition, we also add another experiment in Sect 5.3 regarding the relationship between hyperparameter $\lambda$, which controls the level of uncertainty, and accuracy/robustness. We consider the above experiments to be studying the correlation between uncertainty and robustness.
>
> Since different people may have different personal viewpoints, we are looking forward to hearing from you about your opinion on what is a **true** study of uncertainty in deep learning, and what is a **true** study of uncertainty and robustness.
>
> 2. You pointed out that **"Label smoothing alone does not provide stronger adversarial robustness" is pointed out by many existing works.**
> We would like to show our evidence that, whether label smoothing provides stronger robustness is still an unsettled question.
> An ICLR 2019 submission (**Label Smoothing and Logit Squeezing: A Replacement for Adversarial Training?**, https://openreview.net/forum?id=BJlr0j0ctX), which claims that label smoothing can achieve robustness better than adversarial training, received 25 replies, most of which on experiments. The authors provide experiments in the replies on # steps, # restarts, varying step sizes, and unbounded eps, but still, the doubts are not settled, and the authors eventually withdrew their submission.
>
> Also, whether entropy maximization can provide stronger adversarial robustness is also an unsettled problem. The paper **Improving adversarial robustness via promoting ensemble diversity**, ICML 2019 uses the technique entropy maximization, and claims that it achieves stronger robustness. However, their experiments are only done on PGD10, which is not strong enough and far from conclusive.
>
> We, therefore, consider whether entropy maximization and label smoothing alone contribute to stronger robustness as unsettled problems. Since you said that "LS cannot provide stronger robustness, which is pointed out by **many** existing works", but did not provide **any** references, we look forward to your clarification on your opinion.
>
> 3. You pointed out that "Combining label smoothing/EntM with adversarial training can further provide stronger adversarial robustness", this is the main conclusion of this work. However, I am not fully convinced this is novel enough as this has been observed by existing work. " However, as far as we know, there are no previous works that study the combination of EntM and AT, while our paper carries out an extensive experimental evaluation on the method of EntM. Also, we provide a margin/gradient norm analysis in Section 6, which explains the result and cannot be seen in previous works. Therefore, we think that our contribution is adequate and distinct from existing works.
>
> 4. **Regarding your concerns about different types of attacks,** we added experiments using CW (margin) attacks in the new revision of this paper, see Table 1 and 2. Under the CW (margin) attacks, PAT-EntM also outperforms PAT, and is comparable or more robust than TRADES. Therefore, we consider that our claim and contribution is valid.
>
> We look forward to clarifications from you.

---

### Official Review · AnonReviewer1 · 2020-10-27
**Review of "Rethinking Uncertainty in Deep Learning: Whether and How it Improves Robustness "**

**Rating:** 6
**Confidence:** 2

**Review:**

# Summary
This paper investigates the complementary mechanisms of adversarial training and uncertainty promoting regularizers. In the field of adversarial machine learning, adversarial training as proposed by Madry et al. 2017 has been the common method. In the field of uncertainty regularization, maximum entropy and label smoothing have been the accepted methods. However, the combination of both has not been investigated before. The paper provides extensive experiments on these methods. The final section gives insights in the theory behind adversarial training and uncertainty promoting regularization, where they show that the combined method increases a notion of normalized margin and a notion of adversarial robustness.

# Strong & Weak points

## Strong points

  * The related work section contains an extensive overview of related and contemporary literature, where both literature in adversarial ML and uncertainty promotion is being discussed.
  * Ablation experiments in figure 1 and 2 show that combining adversarial training and uncertainty promoting regularizers have better accuracy under a range of attack settings.
  * Theoretical insights as to why Entropy Maximization would help improve adversarial robustness, complementary to adversarial training is provided in Section 6

## Weak points

  * The method is studied on small datasets (CIFAR, MNIST, SVHN) using small models (ResNet18). It remains to be seen how these insights translate to larger datasets (ImageNet) and larger models (ResNet50).
  * In the abstract and introduction, the text speaks of “true” robustness, against “strong” attacks, but these terms are never defined.
  * The increase in (approximate) normalized margin when using entropy maximization as shown in table 4 provides an important argument for the claim of this paper. However, the numbers were obtained using the CIFAR10 dataset and a ResNet18 model. A stronger case could be made with a larger dataset and a larger model.

# Statement

Recommendation: 6

Reasons
   * The claims for a complementary benefit of adversarial training and uncertainty promoting regularization are backed up with both extensive experiments and theoretical insights.
   * The fields of adversarial training and uncertainty regularizers have been evolving separately and this paper provides initial insights how these two lines of research can be combined. [Comment: I am not 100% up to date with the related literature, so I'll be looking to other reviewers if they are aware of existing work combining uncertainty regularization and adversarial training.]
   * Ablation experiments in Figure 3 shows substrates for the complementary actions of Entropy Maximization and Adversarial training. Entropy maximization can be shown to increase a notion of “normalized margin width”, adversarial training can be shown to increase a notion of  adversarial robustness, and when combining the methods, both metrics increase.

# Additional questions:

Table 3a) misses the point cloud for normal training. How do the normalized margins and adversarial distances of normal training compare in this plot? Table 4 already shows that the normalized margin (0.19) is smaller than the three methods, but I miss the numbers for adversarial distances during normal training.

# Minor feedback

These points are minor feedback and not part of the assessment.

  * KL divergence can be decomposed as $KL(p|q) = H[q] + CE(p|q)$. Could this decomposition explain the differences between TRADES, PAT, and entropy maximization?

  * Figure 1 & 2: please provide labels for the x axes.
  * None of the derivations on eqn. 9 to 13 depend on $\theta$. Consider dropping the $\theta$ variables everywhere to focus the analysis on what really matters: the derivatives w.r.t. $x$.
  * Equation 12: the $\delta$ variable is only implicitly defined. It is not clear to me if its direction is parallel or orthogonal to the decision boundary.

---

> ### Author Response · Authors · 2020-11-13
> **Thanks for your comments. Response.**
>
> We are grateful to your helpful comments to our paper.
>
> We address some of your comments below.
> 1. **Regarding larger datasets (ImageNet) and larger models**.
> Regarding dataset, in fact, many papers in the field of adversarial robustness fail to carry out experiments on Imagenet. The reason is probably that Imagenet is too resource consuming. As pointed out by https://arxiv.org/pdf/1904.12843.pdf, footnote 1, adversarial training on imagenet may take hundreds of GPUs like Tesla V100 and weeks of time.
>
> Regarding larger models, we point out that ResNet18s are commonly used in the field of adversarial training, such as TRADES (Zhang et al. 2019), and the paper "Overfitting in adversarially robust deep learning" (Wong et al. 2020). Therefore we consider ResNet18 to be also appropriate. To be aligned with the SOTAs, we are running experiments on Wide-resnet32s, which are also commonly used in the field of adversarial robustness and bigger than ResNet18. We will update our paper by then.
>
> 2.**Regarding point clouds of Normal**.
> We once added the Normal into the point cloud, but the points of Normal are highly cluttered around the 0 point and not clearly visible. Therefore we omit the points by "Normal". We think that the Normal points will not affect our claim and our results.
>
> 3. **Regarding terms strong adversary and true robustness**.
> We admit failing to clearly define them. The term "Strong adversary" in general refers to attacks with more effort, but under the same budget. For example, we can say that PGD10-4 is stronger than FGSM4, and PGD40-4 is stronger than PGD10-4, but we generally do not say that PGD10-8 is stronger than PGD10-4 (because they are given different budgets). However, the term "strong attack" is a relative term, and is hard to be rigorously defined.
>
> The term "true robustness" represents the same meaning as the term "adversarial risk" defined in (Uesato et al. 2018). We will revise our paper and replace all "true robustness" by "adversarial risk". In short, adversarial risk is a term that measures the "existence" of adversarial examples around a data point, regardless of specific attacks. It is a term that upper bounds the success rate of all attacks, and cannot be measured but can only be approximated by stronger and stronger attacks.
>
> 4. **KL divergence**
> The KL divergence is used in our paper to show the connection between entropy maximization and label smoothing. See Eqn. 7-8. However, since KL(p|q) = H(q) + CE(p|q), the entropy term is taken over q, i.e. labels, which is not directly related to the outputs of networks. We leave further investigation as future work.
>
> 5. **Fig. 1, 2 labels**: We have updated them in the latest revision.
>
> 6. **Dropping $\theta$**. The suggestion is highly helpful and we have dropped it.
>
> 7. **$\delta$ variable**. We admit failing to state Theorem 1 very clearly. $\delta$ is an arbitrary noise, and as long as the condition in Eqn. 12 holds, the $\delta$ direction can be arbitrary. We updated the descriptions of Theorem 1 to make things clearer.

---

### Official Review · AnonReviewer3 · 2020-10-27
**Need Novelty Clarification**

**Rating:** 5
**Confidence:** 4

**Review:**

Summary:

This paper tries to improve the model robustness by modifying the loss function with the EntM or LS term. Also, they give a further analysis to identify how uncertainty promotion works.


Strength:

-- The methods seem rational. And the experiments also demonstrate the effectiveness.


Weakness:

--The novelty is limited. Training the model without the one-hot label is not new. Also, it seems that the label smoothing is the only contribution proposed by this work, but it is pretty naive.

--The analysis part is difficult to follow.


Comments:
--There are many works that claim the distillation training can improve the robustness of the model, which also adopt the soft label in loss function. Please compare with relative methods.

-- For uncertainty analysis, the explanation is not straightforward. First, I am not sure why Eq. (11) holds. Second, why the L2 norm of the Jacobian matrix is its largest singular value? Also, I found the analysis is finally conducted through empirical studies. I think maybe the equation descriptions can be modified easier to understand and make the empirical study more clearly if possible.

-- I think the performance drop with adversarial training is still high, although the results show the accuracy is better than TRADE. I suggest the authors to provide some experiments that adjust the hyper parameters to identify the trade-off between the clean accuracy and the robustness of the model.

--For Eqs. (7)-(8), what does the term ‘logC’ stand for?

--This work only considers untarget attack. How about the performance under target attack?

--The attack algorithms seem not state-of-the-art. Please try more algorithms if possible.

---

> ### Author Response · Authors · 2020-11-13
> **Thanks for your comments. Reply.**
>
> Thanks for your comments. We quickly reply to your doubts on the paper.
>
> 1. **Novelty**. Although the method of entropy maximization and label smoothing is not new, we believe that our paper still provides contributions that are different from existing works. First, we show that EntM and LS alone can only provide partial and conditional robustness (under weak attacks, or under attacks with low perturbation), and that EntM and LS alone cause gradient obfuscation. Previously, as far as we know, the question of how robust EntM and LS alone are is not settled. Please refer to 2. in the reply to Reviewer 4 for details.
>
> Also, we demonstrate through extensive experiments that EntM and LS consistently improves PAT over all perturbations and a wide range of attacks, which is contrary to the case of EntM and LS alone. We also provide an analysis of why this happens in Sect. 6, which, as far as we know is not done by previous works. Therefore, although we agree that the method is not new, we still make adequate contributions that are not solved by existing works.
>
> 2. **The analysis part is difficult to follow.** Including Eq 11 and Norm of Jacobian Matrix.
> We agree that the analysis part is not described clearly, and we have added related discussions in the updated paper.
> Eq 11 is a direct outcome of Eq 10, which basically divides a positive term $\|X'-X\|_2$ on both sides and re-arranging the LHS and the RHS.
> For a matrix $M$, in general we use $\|M\|_2$ to denote its spectral norm, which is defined to be its largest singular vector. (See https://en.wikipedia.org/wiki/Matrix_norm).
> Also we will modify the descriptions to make them clearer.
>
> 3. **Regarding hyperparameter experiments and the tradeoff**
> We have done hyper parameter experiments in Sect. 5.3 and Fig. 3. The results show that EntM can achieve improvements on both accuracy and robustness, thus partially alleviating the tradeoff.
>
> 4. Eq 7, 8, **C** stands for the number of classes, which is defined in 3 Preliminaries.
>
> 5. **Targeted attacks.** Since our defense does not focus on a specific "victim" class to strengthen, and untargeted attacks are more difficult to defend than targeted ones, we think that untargeted attacks would be sufficient to demonstrate the effectiveness.
>
> 6. **Attacks not SOTA.** We added results using CW attacks to attack our defenses (See Table 1 and 2). The results are consistent with using PGD and do not contradict our claims.
>
> 7. **Distillation Training**. We note that a work in AAAI 2020, **Adversarially robust distillation** (ARD, https://arxiv.org/abs/1905.09747), mentioned the problem of distilling a teacher model into a student model while maintaining robustness. We summarize several findings made by that paper to answer your questions.
> - First, ARD aims to build small but robust models from large models without adversarial training, which is different from our focus. Our focus and contribution do not rely on existing large models, and aim to build better adversarially robust models from scratch.
> - Second, ARD shows that non-robust teachers lead to non-robust students (See Table 2 of ARD). Therefore, in order to obtain a robust student model, a robust teacher should be obtained, probably via AT. In this way, in terms of time consumption, ARD is no more efficient than ours.
> - Third, ARD shows that through ordinary distillation procedures, student models distilled from teacher models show performance drops in terms of both accuracy and robustness. See Table 3 of ARD.
> - Last, ARD shows that using the proposed ARD technique, robust teachers can be distilled to students with robustness preserved. However, ARD requires another procedure of PGD-AT, which takes more time than our approach.
>
> We will add the above discussions to the Related Work section of our paper.

---

### Official Review · AnonReviewer2 · 2020-10-30
**Interesting combination of ideas to increase adversarial robustness but gaps in the experimental evaluation**

**Rating:** 5
**Confidence:** 4

**Review:**

The authors combine adversarial training with two methods that increase the entropy of the output distribution of neural networks (label smoothing and entropy maximization). The authors find that this combination of ideas increases adversarial robustness on standard benchmarks, especially in the regime of large perturbation budgets (e.g., 16/255 on CIFAR-10). The authors also investigate the effect of their methods on the classification margin to understand the increase in adversarial accuracy.

While the authors provide a detailed experimental evaluation of their defense method, this part of the paper is still my main concern. In particular, I see the following issues:

* It is not clear if the authors choose a sufficiently large steps size for the PGD attacks. Typical values are 2 * eps / k to 10 * eps / k, where eps is the l_inf perturbation budget and k is the step size. However, the authors choose smaller step sizes in at least some cases, e.g., eps / k for the perturbation accuracy curves in Figure 1. This is a concern particularly because Table 1 shows that the model accuracies still decrease substantially when going from 10 to 40 PGD steps.

* Again on the note of step sizes, why did the authors choose max(1 / 510, eps / k) in the attacks with more iterations?

* It would be good to present a single table with the smallest known adversarial accuracies for the various defenses. E.g., Table 1 does not seem to include the results from more adaptive attacks where TRADES is sometimes comparable to the proposed methods.

* How did the authors choose random restarts? They mention sigma = 0.005 - is this for a Gaussian distribution? What happens if the authors pick a point from the appropriately scaled l_inf ball instead?

* It would be good to see attacks with the Carlini-Wagner (margin) loss function.

Considering the well-known difficulties with evaluating defenses against adversarial attacks, I currently cannot recommend accepting the paper.

Additional comments:

- The authors attribue Szegedy et al., 2013 with adversarial examples. The authors may be interested in https://arxiv.org/abs/1712.03141 , in particular Figure 8, for a more detailed history of this research direction.

- Trade-offs between adversarial and standard accuracy have been studied before Rice et al., 2020. For instance, see https://arxiv.org/abs/1805.12152 and https://arxiv.org/abs/2002.10716 .

- Could scaling the softmax temperature also work for increasing the entropy of the softmax distribution in a way that leads to increased robustness?

- Could the proposed technique be combined with TRADES or training on additional unlabeled data? (e.g., see https://arxiv.org/abs/1905.13736 and https://arxiv.org/abs/1905.13725 ).

- Is there a "max" missing in Equation 5?

- Beginning of Section 6: "deeper into the how"

- Equation 12: what is X + delta?

---------------------------------------------------------------------------------------------

Thank you for the detailed response. I have updated my score from 4 to 5.

---

> ### Author Response · Authors · 2020-11-12
> **Thanks for your helpful comments regarding experimental evaluations. New experiments done.**
>
> We are grateful for your comments regarding deficiencies in experiments. We do additional experiments to address your concerns. Revisions of the paper has been uploaded.
>
> 1. **Regarding small step size.** Upon your comments, we redo the experiments in Fig. 1 with step size 2eps / #steps, or in the experiments, stepsize = 2/255, with eps k/255 and k steps. The curves are roughly the same as using eps / #steps, but generally lower accuracies are observed.
> Also, for Table 1, 2, 3 experiments, we use eps = k/255, but take step size **k/2550 regardless of # steps**. This means that for 40-step attacks, step size = 4eps / # steps, which should be sufficient. Therefore, the results in Table 1, 2, 3 under PGD40 should be credible.
>
> 2. **Also regarding step size choice of max(1/510, eps/k).**
> Upon your comments, we redo the experiments in Fig. 2 (attacks with more steps) with step size 2eps / k. We find out that by using 2eps/k, **attack success rates converge earlier than using max(eps/k, 1/510), and to a lower success rate**. We therefore use a constant step size (1/510) when k is very large (when max(2eps/k, 1/510) takes 1/510), which leads to more powerful attacks. This result shows the importance of using a **constant (rather than decaying with k)** step size when k is large.
>
> 3. **Regarding worst-case robustness**.
> We made a revision to the paper, which explicitly states the worst-case robustness of PAT-EntM under CIFAR-10, eps = 8 (See Sect 5.2.2 Adaptive Attacks). The result is 0.4505 compared to 0.4521 for TRADES, which is still a comparable number. For other scenarios, PAT-EntM is consistently and clearly over TRADES (by at least 2%), and thus we omit another table.
>
> 4. **Regarding random restarts**.
> We use 5 random restarts for each attack, each initialized with Gaussian noise with a standard deviation 0.005.
> Upon your comments, we do some experiments regarding randomness. Gauss 0.005 denotes Gaussian noise with stddev 0.005, and Unif 0.01 denotes uniform noise with range -0.01 to 0.01.
>
> |Initialization    | PAT-EntM, PGD40-8, CIFAR10|
> |--------------------|----------------------------------------|
> |Gauss 0.005     | 0.4618|
> |Gauss 0.001     | 0.4619|
> |Unif 0.005        | 0.4615|
> |Unif 0.01          | 0.4619|
> |Unif 0.02          | 0.4630|
>
> |Num Restarts | PAT-EntM, PGD40-8, CIFAR10 (Gauss 0.005)|
> |------------------- |--------------------------------------------------------     |
> |1                        | 0.4619                                                                 |
> |2                        | 0.4612                                                                 |
> |5                        | 0.4598                                                                 |
> |10                      | 0.4592                                                                 |
> |20                      | 0.4591                                                                 |
>
> Therefore we claim that randomness does not significantly influence our conclusion. Also, 5 random restarts are sufficient to obtain reliable robustness, and increasing # restarts cannot lead to a much higher success rate.
>
> 5. **CW attacks**
> Upon your comments we did CW margin attacks on CIFAR10 and CIFAR100. The implementation is taken from https://github.com/zjfheart/Friendly-Adversarial-Training/. We list related results in Table 1, 2.
> These results are still consistent with our claim, that PAT-EntM achieves consistent improvement over PAT, and comparable or more robust than TRADES.
>
> - Additional Comments:
> 1. Review Papers: We will add them as more comprehensive surveys than Szegedy 2013 and Goodfellow 2014.
> 2. Adv-Std Accuracy Tradeoff: We have added the papers you mentioned to Sect 5.3, where we discuss one additional experiment done for the revision, and the accuracy-robustness tradeoff.
> 3. Temperature Scaling: We have not done it but it seems unlikely. The reason is that, if we scale the output of a network $f(X)$ by 10, $f_1(X) = f(X)/10$, then both the decision margin $M_{f, X}$ and the gradient norm $\nabla_X f(X;\theta)$ will shrink by 10 times, leading to no change in the normalized margin (Theorem 1 and related discussions). However, in Sect. 6 we can see that EntM effectively enlarges the normalized margin, which marks its difference.
> 4. We listed results of TRADE-EntM in the Appendix. In short, TRADES-EntM is better than TRADES, but not better than PAT-EntM. We consider unlabeled data as future work.
> 5. There should be a "max". We fix all typos.
> 6. $\delta$ is an arbitrary noise. We have added descriptions.

---

### Author Response · Authors · 2020-11-14
**Revision Submitted. New experiments added. Restructure of analysis section.**

Dear reviewers:

We uploaded a revision to our paper according to your comments. The changes are:
1. Addition of CW attack results in Table 1 and 2. Under CW attacks, the results are similar to those under PGD, and do not contradict our claims.
2. Addition of hyperparameter analysis $\lambda$ in Sect. 5.3 and Fig. 3. By choosing appropriate $\lambda$, uncertainty level, we can see both accuracy and robustness improvements.
3. Restructure of the Sect. 6, analysis. We add necessary discussions and descriptions.
4. Discussion of KD methods in Sect. 2 Related Works.
5. Fix minor issues and typos.

Thank you all for your helpful comments.

---

### Author Response · Authors · 2020-11-21
**Revision Submittes. Add experiments of WideResNet34-10.**

Dear reviewers:

We upload another rebuttal revision. The changes are:
1. We add experiments on WideResNet34-10 (which is also commonly used in adversarial learning literature, and used in TRADES). The results are shown in Appendix D.

---

### Decision · Program_Chairs · 2021-01-07
**Final Decision**

**Decision:**

Reject

**Comment:**

The reviews were a bit mixed, with some concerns on the novelty and experimental evaluation. While the authors' efforts during rebuttable were appreciated, the overall sentiment is that this work, in its current form, cannot be accepted to ICLR yet. Please consider revising your work based on the excellent reviews. Some more comments from the AC's independent assessment:

(a) Further elaboration on the novelty is needed. Currently the main message appears to be that if we combine two existing approaches (AT and EntM or LS) then we get better results. This is perhaps not too surprising and more elaboration on the significance would be appreciated.

(b) More comparisons in the experiments, including the SOTA performances and alternative defenses (some below).

(c) The analysis in Section 6 adds more confusion than clarification. It is clear that EntM and LS would largely decrease M_f, but why would they also decrease the Lipschitz constant even more sharply? If this explanation is useful, why not directly regularize the Lipschitz constant and maximize the margin M_f? There is in fact a large body of work on this, see for example:

1. Formal Guarantees on the Robustness of a Classifier against Adversarial Manipulation

2. Parseval Networks: Improving Robustness to Adversarial Examples

3. L2-Nonexpansive Neural Networks

4. and the many references since.